# Compact 8 × 8 MIMO Antenna Design for 5G Terminals

**Haifu Zhang** [1,2], **Li-Xin Guo** [1,*], **Pengfei Wang** [2] **and Hao Lu** [1]

1    School of physics, Xidian University, Xi'an 710071, China
2    54th Research Institute of CETC, Shijiazhuang 050081, China
*    Correspondence: lxguo@xidian.edu.cn

**Abstract:** In this paper, a compact 8 × 8 MIMO antenna design for 5G terminals is proposed. The 8 × 8 MIMO antenna consists of two quad-element antenna pairs, each of which includes two symmetrical T-shaped monopole mode elements and two symmetrical edge-coupled fed dipole mode elements. The size of the quad-element antenna is $38 \times 7 \times 0.8$ mm$^3$. T-shaped monopoles are decoupled by parasitic elements, and dipoles are decoupled by grounding strips. Meanwhile, both T-shaped monopoles and dipoles are also decoupled by the orthogonal mode. The results show that the operating frequency band of each antenna element meets the requirement of 3.4–3.6 GHz, the reflection coefficient is less than −6 dB, and the isolation between any antenna element is more than 10 dB. The antenna radiation efficiency is over 50% in the entire operating frequency band for the 8 × 8 MIMO system.

**Keywords:** multi-input multi-output (MIMO); polarization diversity; compact antenna; parasitic element





## 1. Introduction

With the continuous improvement of mobile communication quality requirements, the fifth generation (5G) mobile communication technology provides a promising solution for high communication rate, low latency, large connection density and high communication capacity. In order to meet the goals of 5G mobile communication and effectively improve the channel capacity of the communication system in a rich scattering environment, MIMO technology has become a key technology in the new generation of wireless communication systems [1]. Generally, the MIMO systems enlarge the channel capacity through multiple independently placed elements, known as spatial diversity. However, due to the narrow space of the terminals, spatial diversity cannot reach its true potential. Thus, other diversity techniques, such as polarization diversity and radiation pattern diversity are applied in MIMO systems. Both the transmitter and the receiver of a MIMO system need to place multiple antennas, and the coupling between the antennas will reduce the performance of the MIMO system [2]. Therefore, the decoupling design of the MIMO antenna has become an important part of the wireless communication system. With the increase in the number of antennas on smart devices and the miniaturization of terminal equipment, the space left for antenna placement is limited. How to reduce mutual coupling between antennas and achieve high isolation in a compact space is a difficult problem.

Traditional decoupling methods, such as neutralization line [3–5], parasitic element [6–8], EBG electromagnetic band gap [9–11], DGS defect ground structure [12–15] and decoupling network technology [16–18], all need to add additional decoupling structures in the middle of the antenna, which increases the complexity of the design. The occupied area of the antenna is large, and it cannot meet the compact requirements of the future 5G terminal antenna.

Most of the initial polarization diversity technologies [19–22] use the same antenna elements placed orthogonally to achieve decoupling, which undoubtedly increases the occupied area of the antenna. Therefore, researchers use different antenna elements to

achieve polarization diversity through their orthogonal current modes, which enhances the isolation and improves the compactness of the antenna.

In this paper, the polarization diversity is combined with the parasitic element to reduce the isolation of the designed compact MIMO antenna. The antenna integrates two T-shaped monopole elements and two edge-coupled dipole mode elements on an area of $28.6 \times 7$ mm$^2$, which greatly improves the integration degree of the antenna design. The orthogonal current modes of two types of elements enhance the isolation. In addition, parasitic elements and grounding strips are used to weaken coupling between elements of the same type. The operating frequency band of each element is 3.4–3.6 GHz, with an S$_{11}$ less than $-6$ dB. The isolation between any elements is above 10 dB. The simulation was performed with the help of Ansys HFSS [23]. This article is organized as follows. Section 2 describes the antenna configuration and decoupling principles. Section 3 presents the experimental results and discussion. Finally, Section 4 is the conclusion of this paper.

## 2. Antenna Configuration and Decoupling Principle

Figure 1 shows the overall structure and detailed dimensions of the eight-element highly integrated antenna. As shown in Figure 1, the antenna consists of two 4-element highly integrated antenna pairs, and each antenna pair consists of four antenna elements: two symmetrical T-shaped monopole-mode antenna elements and two symmetrical edge-coupled-fed dipole-mode antenna elements, printed on the inner and outer sides of the side plates, respectively. The material of the substrate is FR4. The size of the substrate is $150 \times 75 \times 0.8$ mm$^3$, and the area of the ground is $150 \times 73$ mm$^2$. The metal ground is printed on the back of the main board, and the width of the clearance area is only 1 mm. Two small FR4 dielectric side plates with a size of $38 \times 7 \times 0.8$ mm$^3$ are installed along the left and right sides of the ground plane.

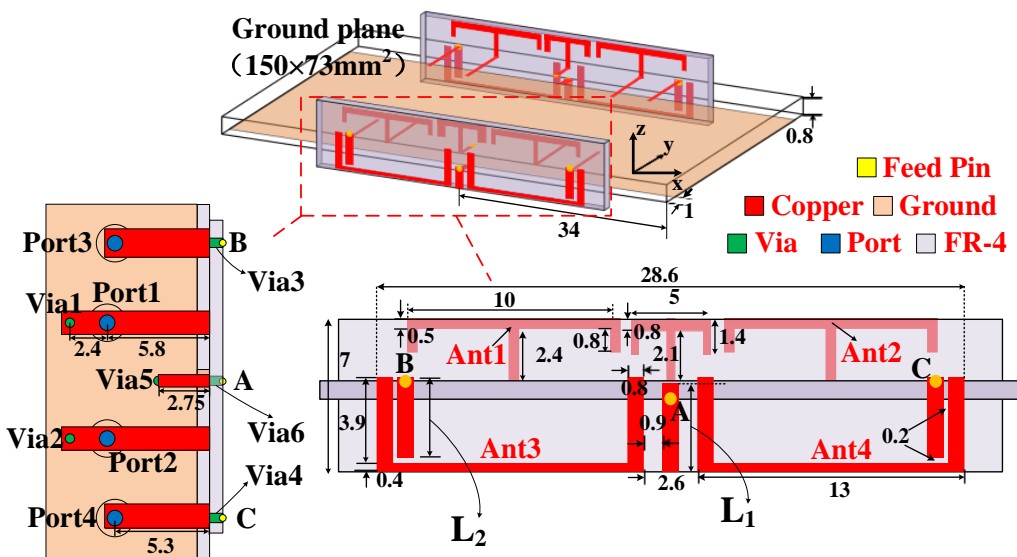

**Figure 1.** Geometry of the proposed MIMO antenna array.

The overall occupied area of the compact antenna pair is $28.6 \times 7$ mm$^2$. Ant1 and Ant2 are two symmetrical T-shaped monopole mode antenna elements, and each antenna element occupies an area of $10 \times 2.9$ mm$^2$. A 50 $\Omega$ microstrip line is used to directly feed Ant 1 and Ant 2 to form a monopole current mode. Microstrip lines with a length of 2.4 mm are extended at port 1 and port 2, respectively. They are connected to the metal ground through the metallized Via1 and Via2 to adjust the operating band of the monopole mode antenna element while improving the impedance matching. A T-type parasitic element is introduced between Ant1 and Ant2. The structure is connected to the metal ground through a microstrip line and Via5. The decoupling of Ant1 and Ant2 is mainly realized through the T-type parasitic element structure.

Ant3 and Ant4 adopt the form of edge-coupling feed to realize the dipole current mode, and each antenna element occupies an area of $13 \times 4.3$ mm$^2$. As shown in Figure 1, a 50 Ω microstrip feed line is connected to the coupled feed branch on the side plate through Via3 and Via4 at points B and C. The decoupling of Ant3 and Ant4 is mainly realized by the ground strip between them. The strip is connected to the metal ground through Via6 on the side plate at point A.

The optimized parameters are as follows: $L_1 = 3.7$ mm, $L_2 = 3.7$ mm.

### 2.1. Decoupling Principle between Ant1 and Ant3

Figure 2 shows the current distribution on the antenna and ground when port1 and port3 are excited at 3.5 GHz, respectively. As shown in Figure 2a,b, when the monopole (Ant1) mode is excited, the current on the antenna element is distributed in the same direction along the Z-axis. The current distribution along the X-axis is in the opposite direction, and there is a radiation zero point at the center.

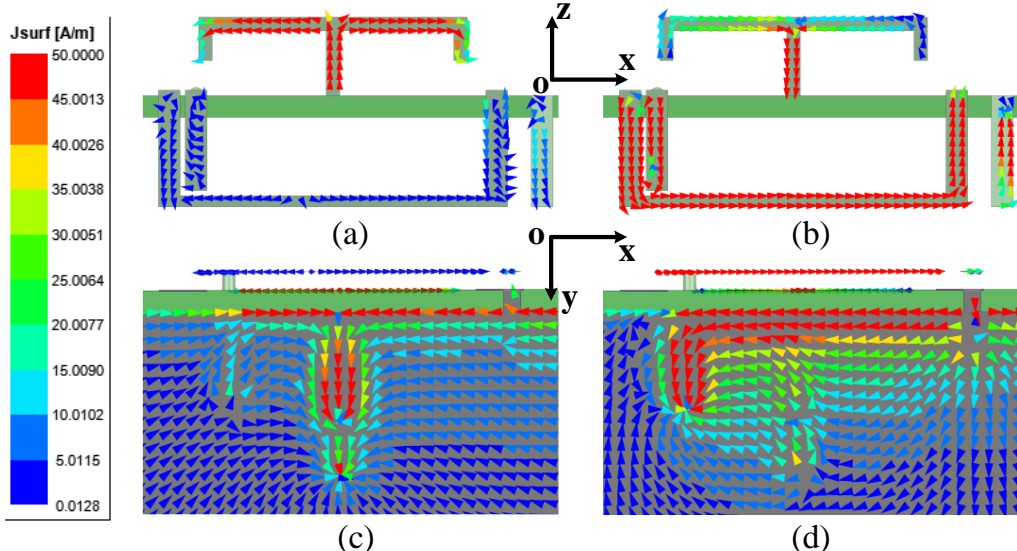

**Figure 2.** (**a**) Current distribution on the antenna when Port1 is excited at 3.5 GHz (monopole mode). (**b**) Current distribution on the antenna when Port3 is excited at 3.5 GHz (dipole mode). (**c**) Current distribution on the ground when Port1 is excited at 3.5 GHz (monopole mode). (**d**) Current distribution on the ground when Port3 is excited at 3.5 GHz (dipole mode).

When the dipole (Ant3) mode is excited, the current on the antenna is distributed in the same direction along the X-axis, and the radiation is the strongest at the center position. In this way, orthogonal antenna current modes are formed, so that there is no mutual coupling in the space between the antennas. There is also a set of quadrature modes, which are the orthogonal currents on the ground. As shown in Figure 2c,d, the currents on the ground are also orthogonal, so the current coupled through the ground is also blocked.

For the edge-fed dipole (Ant3), the grounded strip is the key to decoupling not only the two edge-fed dipoles Ant3 and Ant4, but also the monopole Ant1 and the edge-fed dipole Ant3. If the strip is not grounded, it will affect the current balance; the isolation between the two antenna elements Ant1 and Ant3 will deteriorate rapidly. As shown in Figure 3, if the grounded point of Ant3 is removed, the isolation between Ant1 and Ant3 will rapidly deteriorate to below 10 dB. It can be seen in Figure 3 that, through the introduction of Via6, the performances of $S_{11}$ and $S_{33}$ are somewhat worse, but these still meet the requirement of −6 dB.

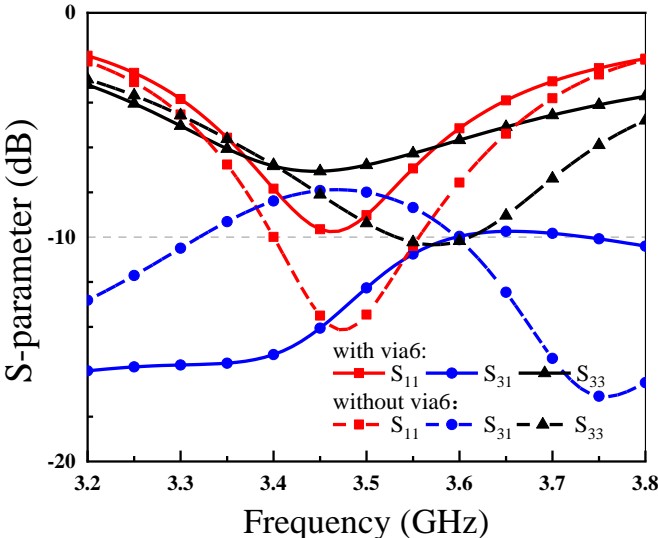

**Figure 3.** Simulated S-parameter of Ant1 and Ant3 with and without Via6.

### 2.2. Decoupling Principle between Ant1 and Ant2

Figure 4 shows the current distribution on Ant1 and Ant2 with or without the grounded pin Via5 when Ant1 is excited. It can be clearly seen that when Via5 is provided, the T-type parasitic element can effectively excite the current to form a current loop with Ant1 and block the coupling current from entering Ant2.

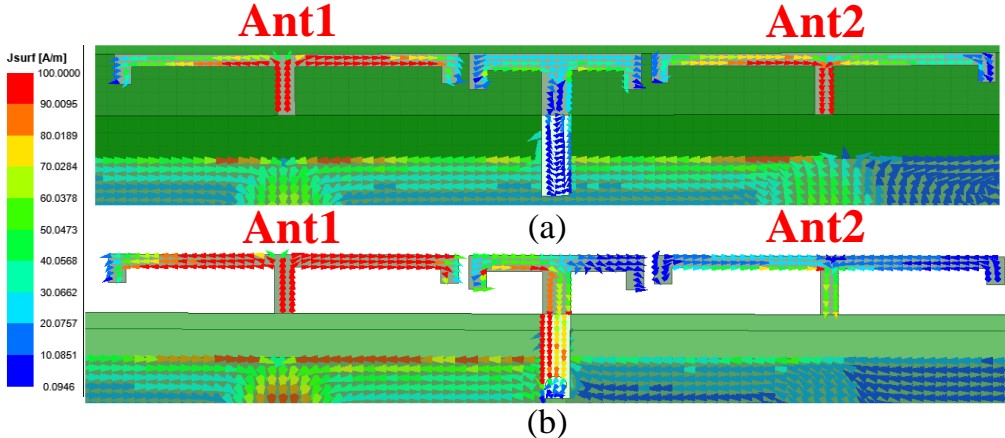

**Figure 4.** (**a**) Current distribution when Ant1 is excited without Via5. (**b**) Current distribution when Ant1 is excited with Via5.

In order to describe the decoupling mechanism more clearly, the current model diagram is established as shown in Figure 5. In Figures 4 and 5, it can be seen that when there is no Via5, the space coupling current forms a current loop between the two antenna elements, and the T-shaped decoupling branch in the middle cannot effectively excite the current. When there is a grounded pin Via5, the half side of the T-type decoupling branch close to Ant1 can effectively excite the current, forming a current loop with Ant1, and the coupling current intensity on Ant2 is significantly reduced. Figure 6 shows the simulated S-parameter of Ant1 and Ant2 with or without Via5. It can be seen that when Via5 is added, the isolation between antenna elements is significantly improved. However, since the parasitic element will also have a coupling effect on the current on the excitation antenna Ant1, it will affect the impedance matching of the antenna. From the results in Figure 6, the introduction of Via5 enhances the isolation but sacrifices part of the bandwidth of the antenna.

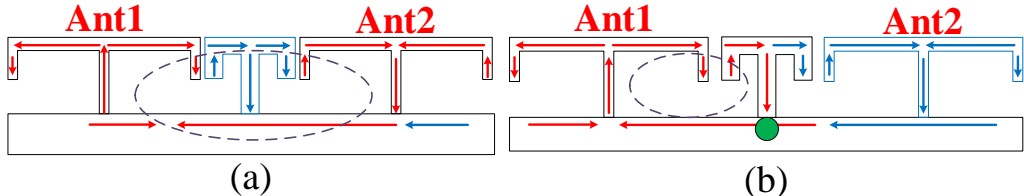

**Figure 5.** (**a**) Current model diagram when Ant1 is excited without Via5. (**b**) Current model diagram when Ant1 is excited with Via5.

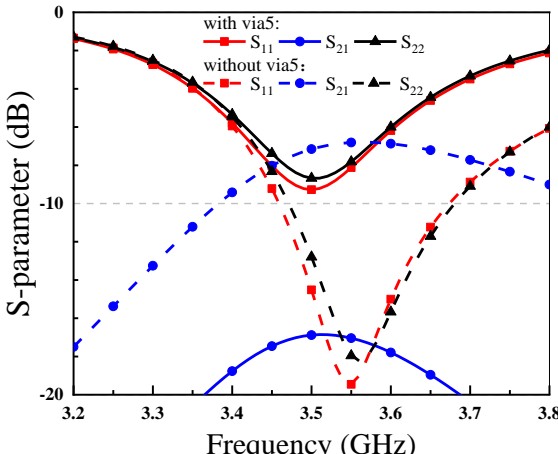

**Figure 6.** Simulated S-parameter of Ant1 and Ant2 with and without Via5.

### 2.3. Decoupling Principle between Ant3 and Ant4

For the edge-fed dipoles Ant3 and Ant4, the grounded strip is the key to decoupling between Ant3 and Ant4, and its length $L_1$ is a key parameter. Figure 7 presents the current distribution at two resonance points, 3.45 GHz (resonance point 1) and 4.1 GHz (resonance point 2), when Ant3 is excited. When there is no grounded strip, that is, $L_1 = 0$ mm, as shown in Figure 7, there are two working modes of Ant3; the resonance point of working mode 1 is 3.45 GHz, and the resonance point of working mode 2 is 4.1 GHz. It can be seen that working mode 1 is not desired, because Ant3 and Ant4 are in mixed working mode at this time. The current loop of this mode is formed by the combination of the current on Ant3 and Ant4 and the current flowing through the ground plane; there must be strong coupling in this working mode. The operating mode 2 is generated by Ant3 and Ant4 separately. Ant3 and Ant4 have separate current loops in this mode, but the isolation of the antenna unit is still poor at this time, because there is a coupling current in the space.

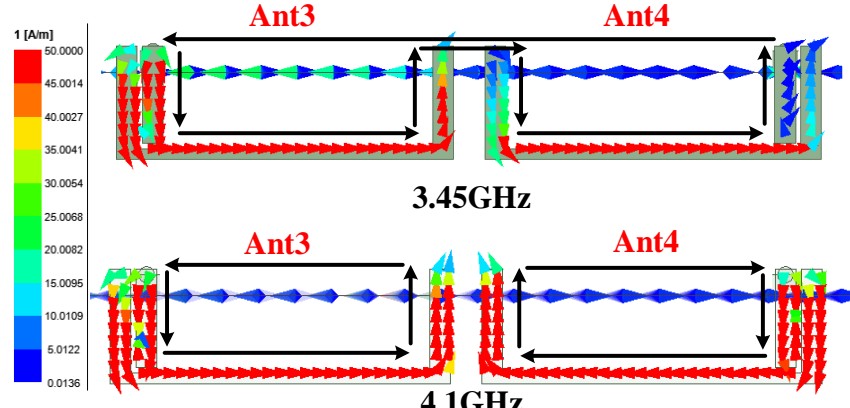

**Figure 7.** Simulated current distributions of Ant3 and Ant4 with port 3 excitation at 3.45 and 4.1 GHz when $L_1 = 0$ mm.

Figure 8 shows the current distribution on the two antenna elements when Ant3 is excited. In Figure 8a, when $L_1$ = 2 mm, the resonance frequency is 3.6 GHz. In Figure 8b, when $L_1$ = 4 mm, the resonance frequency 3.3 GHz. At these two frequency points, the working mode 1 can no longer be effectively excited. The resonance point shifts to the low frequency band with the increase of $L_1$. At the same time, when $L_1$ = 4 mm, only a small amount of energy is coupled into Ant4 at this time, indicating that this working mode can meet isolation requirements when the length of the ground branch is large enough.

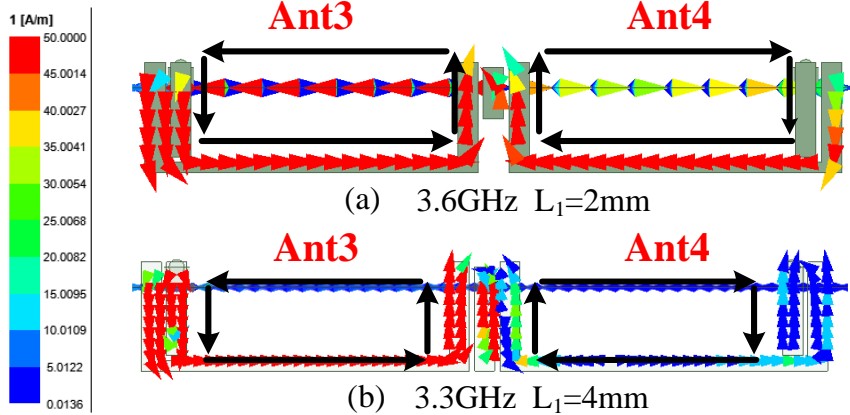

**Figure 8.** Simulated current distributions of Ant3 and Ant4 with port 3 excitation (**a**) at 3.6 GHz when $L_1$ = 2 mm and (**b**) at 3.3 GHz when $L_1$ = 4 mm.

In order to illustrate the role of the parameter $L_1$ more clearly, Figure 9a shows the simulated results of the parameter analysis of $L_1$. It can be seen that with the increase of $L_1$, the electrical size of the antenna increases, and $S_{33}$ shifts to the low frequency band. But the $S_{11}$ increases, sacrificing some bandwidth. When $L_1$ = 3 mm, $S_{43}$ starts to be less than −10 dB.

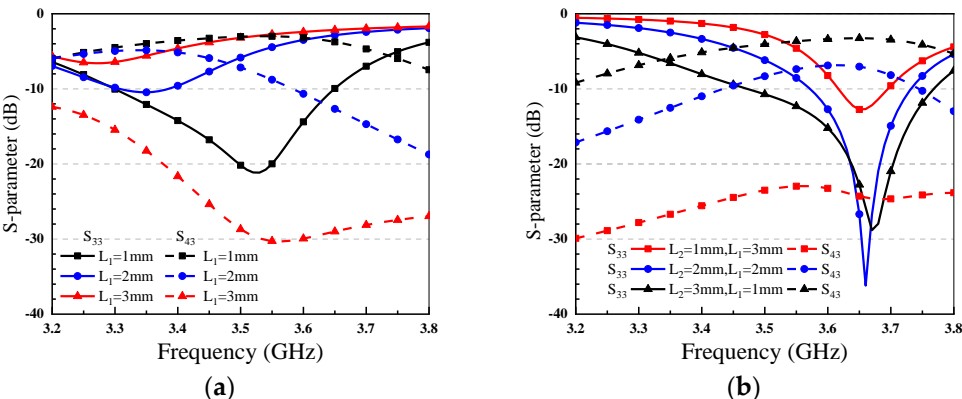

**Figure 9.** (**a**) Simulated S-parameter of Ant3 and Ant4 with different $L_1$. (**b**) Simulated S-parameter of Ant3 and Ant4 with different $L_1$ and $L_2$.

Since $L_1$ affects the matching, $S_{33}$ and $S_{43}$ will change with $L_1$ at the same time. As shown in Figure 9b, with the complementary changes of $L_1$ and $L_2$, the resonant frequency point hardly changes. When $L_1$ = 3 mm, the isolation of the entire frequency band is above 20 dB. Therefore, for Ant3 and Ant4, the length of the $L_1$ strip must be greater than 3 mm, because the isolation can meet the requirements at this time. Then, the length $L_2$ of the coupling feed branch is adjusted to make the antenna work at the desired frequency band.

Table 1 shows the comparison between this work and previous works. Comparing the antenna size and integration, the proposed antenna shows a high compactness and realizes an excellent integration degree, which integrates four elements on an area of only $28.7 \times 7$ mm$^2$.

**Table 1.** Comparison of the proposed antenna pair with previous works.

| Ref. | Antenna Size (mm²) | Integration | Isolation | Bandwidth | ECC |
|------|--------------------|-------------|-----------|-----------|-----|
| [24] | 20 × 7 | Two-element integration | >17 dB | 3.4~3.6 GHz | <0.1 |
| [25] | 12 × 17 | Two-element integration | >20 dB | 3.4~3.6 GHz | <0.06 |
| [26] | 25 × 7 | Two-element integration | >20 dB | 3.4~3.6 GHz | <0.13 |
| [27] | 60 × 5 | Four-element integration | >10 dB | 3.3~7.2 GHz | <0.2 |
| [28] | 38.2 × 3.2 | Four-element integration | >11.8 dB | 3.4~3.6 GHz | / |
| This work | **28.6 × 7** | **Four-element integration** | >10 dB | 3.4~3.6 GHz | <0.35 |

## 3. Results and Discussion

In order to verify the performance of the antenna, the prototype of the antenna is fabricated, as shown in Figure 10. The prototype is measured with a vector network analyzer Agilent E8363B and a microwave chamber. Figure 11 presents the measured reflection coefficient and transmission coefficient. For simplicity, the reflection coefficient and isolation of only one antenna pair are given. The results show that if −6 dB is used as a criterion of reflection coefficient, the bandwidth of the four antenna elements of one antenna pair can cover 3.4–3.6 GHz, and the isolation between any two elements is greater than 10 dB.

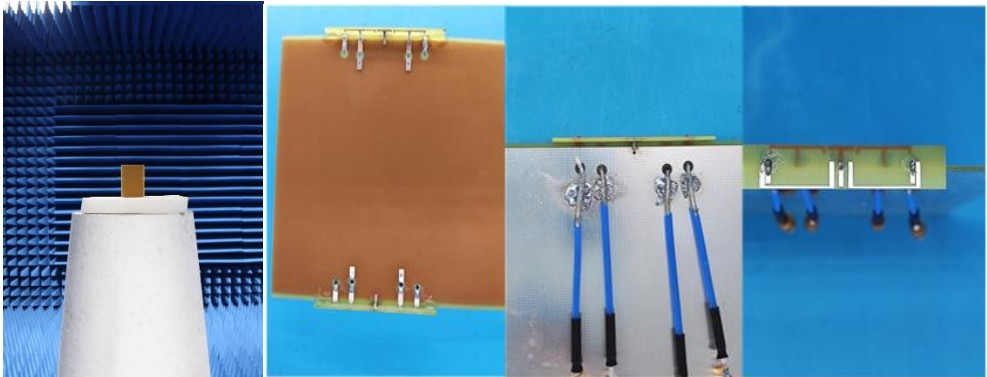

**Figure 10.** The fabricated prototype of the proposed antenna.

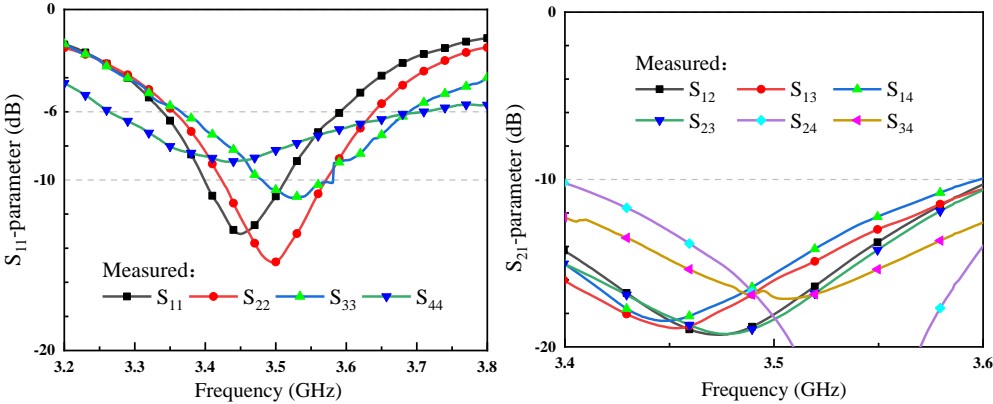

**Figure 11.** Measured reflection coefficient and transmission coefficient.

In order to quantitatively evaluate the diversity performance of the proposed MIMO antenna, the ECC of the antenna is calculated based on the radiated far-field using simulation software; the formula is as follows [29]:

$$E_{cc} = \rho_e = \frac{\left|\oiint[E_1(\theta,\varphi) \cdot E_2(\theta,\varphi)]d\Omega\right|^2}{\oiint|E_1(\theta,\varphi)|_2 d\Omega \cdot \oiint|E_2(\theta,\varphi)|_2 d\Omega} \tag{1}$$

where $E_i\,(\theta,\varphi)$ is the radiation pattern of the antenna when port $i$ is excited.

Diversity gain (DG) is an important parameter to measure good diversity characteristics and MIMO system performance. According to [30], DG can be calculated from Formula (2), and the ideal value of DG should be 10 dB.

$$DG = \sqrt{1 - |\rho_e|^2} \tag{2}$$

It can be seen from Figure 12 that the ECC in the desired frequency band is less than 0.35, indicating that the presented antenna has good diversity performance. It can be seen from Figure 12 that diversity gain values are all around 10 dB, meeting the MIMO diversity criterion. Due to the symmetry of the antenna, only the antenna efficiencies of Ant1 and Ant3 are calculated. As shown in Figure 13, the efficiencies of Ant1 and Ant3 are 52.2–57.8% and 50.2–64.8%, respectively, over the entire operating frequency band.

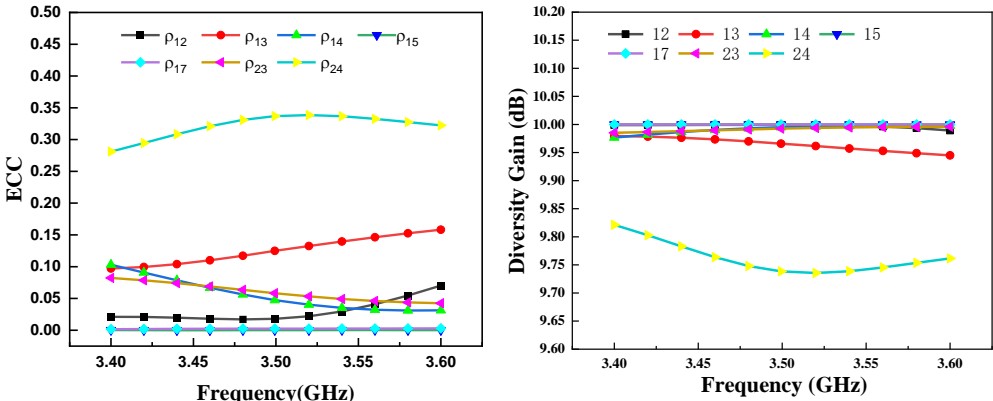

**Figure 12.** ECC and diversity gain of the 8 × 8 MIMO antenna.

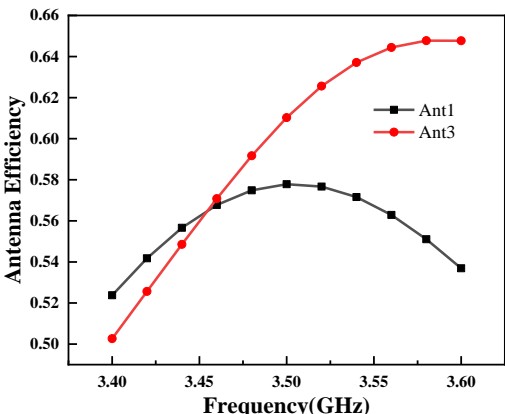

**Figure 13.** The efficiency of the 8 × 8 MIMO antenna.

Figure 14a shows the measured radiation pattern of Ant1 and Ant3 at 3.5 GHz in the XOY plane. As shown in Figure 14a, the radiation zero point of Ant1 is at the center of the antenna, and the radiated energy is concentrated in the positive and negative directions of the X axis, while the Ant3 has the largest radiation energy at the center, but due to the effect of the metal floor, the energy is concentrated in the +Y direction.

Figure 14b shows the measured radiation pattern of Ant1 and Ant3 at 3.5 GHz in the XOZ plane. As shown in Figure 14b, the radiation zero point of Ant1 is at the center of the antenna, and the radiated energy is concentrated in the positive and negative directions of the X-axis; Ant3 has the largest radiation energy at the center, but due to the influence of the metal floor and surrounding radiation patches, the energy is not concentrated in the positive and negative directions of the Z axis, but is shifted. It can be known from

the radiation patterns of Ant1 and Ant3 that the designed antenna has good diversity characteristics. The simulated 3D radiation patterns of Ant1 and Ant3 at 3.5 GHz are presented in Figure 15.

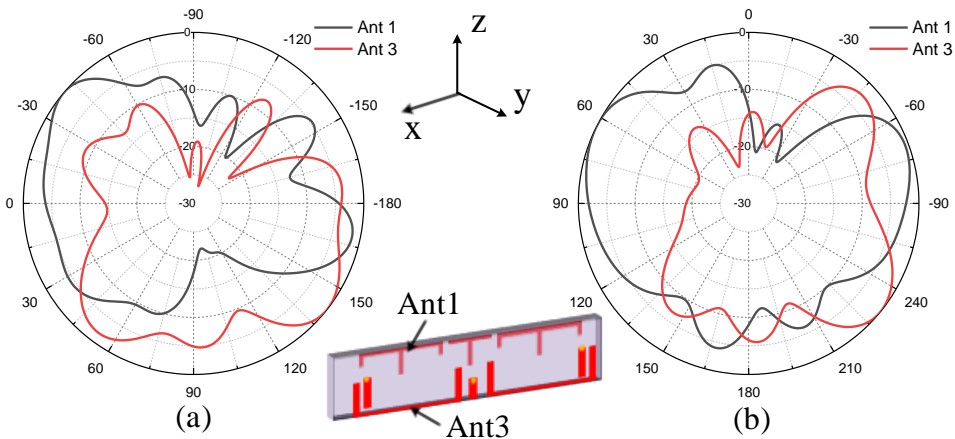

**Figure 14.** Measured radiation patterns of the proposed antenna at 3.5 GHz in (**a**) XOY plane and (**b**) XOZ plane.

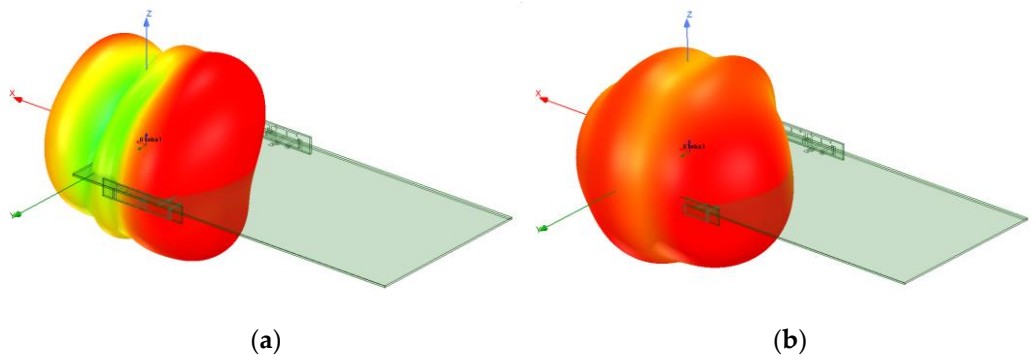

(**a**)        (**b**)

**Figure 15.** Simulated **3D** radiation patterns of the proposed antenna at 3.5 GHz of (**a**) Ant1 and (**b**) Ant3.

## 4. Conclusions

This article presents a compact 8 × 8 MIMO antenna for 5G terminals, which consists of a pair of quad-element antennae. The quad-element antenna is composed of two T-shaped monopoles and two edge-coupled dipoles. The quad-element is placed in a compact area. The antenna adopts polarization diversity technology, parasitic elements and grounding strips to realize decoupling between antenna elements. The measured working frequency band of each element satisfies 3.4–3.6 GHz, and the isolation is above 10 dB. The ECC in the desired frequency band is less than 0.35, and the efficiency is over 50% in the operating bandwidth. The measured results agree well with the simulation. Owing to the convenience and flexibility of the self-decoupled design with a wideband performance, the proposed structure will be widely employed in 5G broadband MIMO antennas. This design provides an important reference for future compact MIMO antenna design.

**Author Contributions:** Conceptualization, H.Z. and H.L.; methodology, H.Z., L.-X.G. and H.L.; software, H.Z. and H.L.; validation, H.Z. and P.W.; formal analysis, H.Z. and P.W.; investigation, H.Z. and P.W.; resources, H.Z.; data curation, H.L.; writing—original draft preparation, H.Z. and H.L.; writing—review and editing, H.Z. and L.-X.G.; visualization, H.Z. and P.W.; supervision, L.-X.G. and H.L.; project administration, L.-X.G. and P.W.; funding acquisition, L.-X.G. and P.W. All authors have read and agreed to the published version of the manuscript.

**Funding:** Project supported by the National Natural Science Foundation of China (Grant No.61871457, No.U21A20457), the Foundation for Innovative Research Groups of the National Natural Science Foundation of China (Grant No.61621005).

**Conflicts of Interest:** The authors declare no conflict of interest.

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
