# Peer review of "Compact 8 × 8 MIMO Antenna Design for 5G Terminals"

_electronics, doi:10.3390/electronics11193245_

Round 1

Reviewer 1 Report

This article presents the design of 8x8 Printed MIMO antenna for 5G with different isolation techniques. I find it an interesting topic, and I have some comments that should be considered before publishing. This paper needs to be organized in a better way.

1-   The abstract needs some improvements. Please add the size of the structure and the values of radiation efficiency.

2-   The introduction section is too short and needs to enhance:- for example the author can write something about the diversity type of the MIMO antenna system as a general and then mention the spatial diversity that he utilized in this paper.

3-   Please point out the contribution of this paper in the last paragraph of the introduction. Usually, this part of the paper gives a comprehensive idea about the main contribution.

4-   Insert a paragraph that explains the details of this paper, Section 2 describes……., Section 3……….. and so on.

5-   Figure 1 in the introduction part while its intext citation in the second section. please rearrange it. The in-text citation of the figure could be before or after the figure while I think it is important to be in the same paragraph no matter which one is the first.

6-   Please mention the type of simulation software in the last paragraph of the introduction and cite it. For instance, the simulation of this work has been performed with the help of “Name of Software” [ref].

7-   There is a typo in the caption of figure 3. It should be “and” not “or” since both results are presented.

8-   Using the decoupling technique via6 between ant3 and ant1 will degrade the return losses above -10 dB. it is worth mentioning that the bandwidth will also affect. Please comment on this point in this paper. and repeat this in any decoupling technique you use.

9-   figure 4 and figure 5 have the same problem as figure 1. they should be in the same paragraph with their in-text citation.

10-   Regarding Table 1. Where is the advantage of this work compared with the literature-reviewed works? Please comment on this in the paper.

11-   Does this work have measurement results of radiation patterns?

12-   The conclusion is too short and needs to improve. For instance, add the results of the efficiency that you got and compare the simulation and measurement results. 

Reviewer 2 Report

Reviews

The topic “Compact 8×8 MIMO Antenna Design for 5G Terminals” presented in the paper using T-shaped monopole and dipole antennas for 5G terminals satisfies the MIMO parameters. But the following comments must be reported.

 ·         In figure 1, the geometry of the proposed MIMO antenna array, the yellow dots (circles) are not highlighted.

·          Are there all four antennas T-shaped in each pair?

·         The axis labels are not same in the figures 3, 6, 9, and 11.

·         There should be curve markers in figures 3, 6, 9, 11, 12, and 13.

·         can the author provide the antenna gain measurement setup, such as an Anechoic chamber, etc.?

·         It is suggested to discuss MIMO performance parameters such as diversity gain, etc.

·         Please correct the dimensions in the comparison table for reference [27].

Reviewer 3 Report

Dear authors,

The proposed work is interesting, but some revisions are necessary to improve the quality of the article. Below are my suggestions:

1. Are the radiation patterns shown measured or simulated? Please, make this clear in the manuscript, and if the results shown are simulated, please add the measured results;

2. Add 3D simulated radiation patterns;

3. Add diversity gain (DG) results;

4. Add ECC and DG results in Table 1;

5. Add a photograph with the measurement setup. In addition, specify the equipment used in the measurement;

6. Considering an application of the proposed antenna in smartphones, add results and analyze the antenna performance (S-parameters and efficiency) in scenarios with the user's hands (single-hand and two-hand modes);

7. In Figure 10, add a photograph of the complete antenna, please.

Round 2

Reviewer 1 Report

The authors did the required corrections and responsed to all my comments. 

Reviewer 3 Report

Dear authors,

Thank you for carefully responding to my suggestions.

In my opinion, the article is ready for publication!